# Keystrokes: A practical exploration of semantic drift in timed word association tasks

**Sean MacNiven**[1,2]*, **Ralph Tench**[1]

**1** Leeds Business School, Leeds Beckett University, Leeds, United Kingdom, **2** Glasgow School of Business for Society, Glasgow Caledonian University, Glasgow, United Kingdom

* S.MacNiven@leedsbeckett.ac.uk

**Data Availability Statement:** The data underlying the results presented in the study are available from OSF: https://osf.io/k9xej.

**Funding:** The author(s) received no specific funding for this work.

## Abstract

This study investigates the phenomena of *semantic drift* through the lenses of *language and situated simulation* (LASS) and the *word frequency effect* (WFE) within a timed word association task. Our primary objectives were to determine whether semantic drift can be identified over the short time (25 seconds) of a free word association task (a predicted corollary of LASS), and whether more frequent terms are generated earlier in the process (as expected due to the WFE). Respondents were provided with five cue words (*tree*, *dog*, *quality*, *plastic* and *love*), and asked to write as many associations as they could. We hypothesized that terms generated later in the task (fourth time quartile, the last 19–25 seconds) would be semantically more distant (cosine similarity) from the cue word than those generated earlier (first quartile, the first 1–7 seconds), indicating semantic drift. Additionally, we explored the WFE by hypothesizing that earlier generated words would be more frequent and less diverse. Utilizing a dataset matched with GloVe 300B word embeddings, BERT and WordNet synsets, we analysed semantic distances among 1569 unique term pairs for all cue words across time. Our results supported the presence of semantic drift, with significant evidence of within-participant, semantic drift from the first to fourth time (LASS) and frequency (WFE) quartiles. In terms of the WFE, we observed a notable decrease in the diversity of terms generated earlier in the task, while more unique terms (greater diversity and relative uniqueness) were generated in the 4th time quartile, aligning with our hypothesis that more frequently used words dominate early stages of a word association task. We also found that the size of effects varied substantially across cues, suggesting that some cues might invoke stronger and more idiosyncratic situated simulations. Theoretically, our study contributes to the understanding of LASS and the WFE. It suggests that *semantic drift* might serve as a scalable indicator of the invocation of language versus simulation systems in LASS and might also be used to explore cognition within word association tasks more generally. The findings also add a temporal and relational dimension to the WFE. Practically, our research highlights the utility of word association tasks in understanding semantic drift and the diffusion of word usage over a sub-minute task, arguably the shortest practically feasible timeframe, offering a scalable method to explore group and individual changes in semantic relationships, whether via the targeted diffusion of influence in a marketing campaign, or seeking to understand differences in cognition more generally. Possible practical uses and opportunities for future research are discussed.

**Competing interests:** The authors have declared that no competing interests exist.

## Introduction

The Belgian surrealist painter René Magritte once added the now famous phrase "Ceci n'est pas une pipe" (this is not a pipe) to his painting (of a pipe) entitled "The Treachery of Images" ("La Trahison des Images", 1929). When asked why he had written it, he replied "The famous pipe. How people reproached me for it! And yet, could you stuff my pipe? No, it's just a representation, is it not? So if I had written on my picture 'This is a pipe', I'd have been lying!" [1]. Magritte recognized that while presentation and representation were clearly and rationally very different, in cognition the distinction often seemed diminishingly small, as we not only process reality, but dynamically generate and regenerate versions of it [2].

Not surprisingly, part of being human involves sensemaking, and many theories have arisen seeking to explore the various intricacies of human cognition, with a prominent subset positing a dual model of cognition such as Kahneman's famous theory of *System 1* and *System 2* thinking, whereby system 1 is fast and system 2 slow and deliberate [3]. Similar to Kahneman's model, the *Elaboration Likelihood Model* posits a dual processing path, suggesting the likelihood to engage with a persuasive argument depends on *central* and *peripheral route* processing, based on factors such as personal relevance [4]. Similar theories of language and cognition has also made their way into the literature. The core assumption of Paivio's Dual Coding Theory (DCT) is that cognition is the result of the interaction of two multimodal systems, that while independent, are interconnected; a verbal system, and a non-verbal or imagery system for processing information such as images and sounds [5,6]. In this model, different cues may elicit different responses and use different systems, with concrete concepts purportedly easier to invoke and remember [7,8]. Somewhat narrower in scope than DCT, and a focus of the present study due to its relevance to language processing and conceptual knowledge rather than cognition more generally, is Barsalou's Language and Situated Simulation (LASS) theory. LASS emphasizes the role of situated simulations (recalled experiences or imagination) in terms of language comprehension [9]. In this interpretation, the duality of the processing models includes a language system, as well as a model simulation system, whereby the brain (re) constructs objects and concepts in context dependent representations, often re-enacting whole scenarios or states [9]. LASS is grounded in how language and experience are intertwined, suggesting that understanding language often requires simulating the experiences that words and phrases denote, making it particularly relevant to semantic processing and comprehension, and, to word association studies [10,11].

Projective, subjective, and intrepid explorers of the unconscious, word associations are in many ways something of a verbal Rorschach test. The study of word associations has a long history, with one of the earliest known studies dating to Galton (1879) and Wundt (1880). The application of word associations to the study of unconscious processes arguably began in earnest with Jung's 1906 paper "Studies in Word Association" [12]. Since then the use of word associations has proliferated, ranging from creativity [13], sustainability [14] and brands [15] to representations for weight loss [16] and mental health [17] and microplastics [18]. Indeed, almost anything you can think of could be explored using word associations techniques, which has led to many studies around fundamental questions of semantic representation, within word association tasks [19–23]. But while words are literally often *carved in stone*, their meaning, the lives they live, and the company they keep over time and context are not, a phenomenon often referred to as *semantic drift* [24].

The present study aims to explore this phenomenon in the context of short, free-word-associations tasks. In this study, we seek to explore to what extent meaning might evolve as we engage, cognitively, with a cue (or idea), and how might the process of imagining and remembering, change how far we veer away from common, established associations to personally

relevant and meaningful simulations? Importantly, can such shifts in meaning be detected in a sub-minute word association task? And if so, how might that inform the design and analysis of future studies that include or even revolve around such free-word-associations?

## Related work

Semantic drift–or its alternative labelling as change, progression, shift or development–has had multidisciplinary treatments from: sociology and anthropology looking at how language is used in society and different cultures and observing dynamic change [25]; psychology where word associations are measured for change over time or how bias is built into cognitive structures for interpreting contexts [26]; computer science where machine learning can be used to measure or predict changes in word meaning and usage or even patterns of change measured over time [27]; then in linguistics scholars have looked at the social factors of language and cultural norms, identity and how different social groups influence semantic change. This can be observed through historical frames by looking at the evolution of language over time [28] and then the changing use of general language through lexicography, sometimes evidenced through dictionary analysis and definitions [29].

There are different studies of semantic change in terms of whether language use is broadening or narrowing as well as observing the causes of change such as sociocultural movements, the phenomenon of metaphorical extension (words extend their meaning over time through diversified usage) and inevitably the role of technology. More recent studies have focussed on semantic change computation in Computational Linguistics [30]. Tang [31] sees this growth as a result of largescale diachronic language data alongside other technological developments to allow for empirical investigation of regularities and mechanisms of semantic change. Similarly, developments in Internet technologies are argued to have accelerated language change [32] as the web encourages new words and their application and usage in web-based communication. Other recent studies have explored semantic drift in context such as observing how language is used and changing as a consequence of expanding interdisciplinary research [33]. Semantic drift they argue is a consequence of diverse disciplinary scholars endeavouring to tackle increasingly complex challenges such as climate change from different academic perspectives. This requires scholars to become familiar with and understand other fields of research and build up and organize disciplinary language and knowledge from multiple fields. This can affect Interdisciplinary Knowledge Retrieval (IKR) and the meaning of concepts can shift.

In terms of studying semantic drift there are studies that use diachronic corpus linguistics, i.e. language corpora using collections of text, speech or writing that come from a specific linguistic field, or computational linguistics which involves the computational modelling of natural language. Others use lexicography to identify semantic changes for example by analysing historical dictionaries.

A large word association study by Laurino et al. [34] found that even in a relatively short time, the company that words keep can change, finding that as compared to the words used before the COVID-19 pandemic, words used during the pandemic led to new associations, such as the word *strain*, which in pre-COVID associations had more frequently been associated with *wine*, while during the pandemic it found itself in the company of decidedly less Bacchic neighbors, including *mutations* and *antibiotics*. Indeed, similar results were found in several other COVID related studies, reinforcing the speed with which semantic drift can erode and reform the semantic landscape [35–37]. But semantic drift is far from limited to large scale, environmental change, and is as normal an occurrence in the blogosphere [38] as it is in many a regular conversation [39]. One aspect to semantic drift that has received little

attention, is the role of the aforementioned cognitive processes themselves on the meaning of words generated via different systems over time, where earlier associations may result from language associations common to a majority of speakers (for example, antonyms, synonyms, hierarchies), while later associations may be the result of conceptual processing via situated generation [11]. Given that multiple systems may be involved in the generation of word associations and that these systems are activated at different times in a word association task [10,11,40], we should expect measurable differences in associates generated over time.

Another important factor expected to influence which words are generated early on in a word association tasks, is the *Word Frequency Effect* (WFE), where high frequency words are generally thought to be both processed and generated more efficiently and faster than low frequency words [41]. Although the WFE does not feature prominently in research applying LASS or dual-coding theories more generally, given that high frequency words are expected to be more strongly grounded in language processes common to speakers and individuals [41–43], while less frequent words may be more closely aligned to sensory or motoric experiences, we expect LASS and WFE to share common traits, two complementary or even mutually inclusive aspects of the same phenomenon.

While language associations, being common and available, should closely align with the WFE, its role may not be as clear for situated simulations (theorized to be largely, but not exclusively episodic, drawing from memory and semantic representation) [9,10]. However, evidence that the first responses to a cue word tend to be very stable and difficult to change (high test-retest reliability) provides some support [44]. Furthermore, in terms of the role of individual and group semantic memory, a recent study by Johns found that "distributional models of semantics have demonstrated the systematic connection between the language that people experience and lexical behavior" [45], with personal variance largely accounted for with even a small amount of aggregation of individually generated terms, ultimately leading to something of a *wisdom of the crowds* effect, again supporting a complex interplay across individual and collective semantic representation.

The WFE is often measured, in terms of the *associative strength*, calculated as how frequently a word is uniquely used in a given context, divided by the number of unique individuals in the group studied [19]. A co-occurrence study employing a 250 character context window around a given cue word of the 1-million word Brown corpus [46] found the "frequency of co-occurrence, corrected for chance, [was] significantly correlated with association strength" [47]. This was further explored by Steyvers et al. [48], in a study of first word associates in a database of over 5000 cue words derived from word association studies, the authors compared latent semantic analyses to a new word association space (WAS), employing associative strength rather than pure co-occurrence. Comparing WAS to LSA in episodic memory tasks, they found that "direct associative strengths [were] the best predictors of intrusion rates in free recall" [48], with WAS achieving substantially higher correlations over LSA.

These findings suggest that word frequency and associate strength are expected to positively influence the speed of recall in a free word association task and are expected to be a key feature of word generation order, and in turn, the likelihood of a word belonging to language associations rather than episodic memory. Connecting the dots from word frequency to LASS, therefore, we posit that high frequency responses to cue words in a word association study will be overrepresented among earlier, *language associations*, while less frequent words may find themselves more frequently in the company of terms generated as part of idiosyncratic *situated simulations*.

Support has also been found for WFE in the brain itself, with one study by Binder et al. [7] using functional Magnetic Resonance Imaging (fMRI) and finding that word frequency not only influences processing efficiency but also affects different brain regions involved in

language processing, where high-frequency words led to reduced activation in regions traditionally associated with language processing, indicating a more streamlined processing path. Exploring the role of simulation in word association, Simon's et al. employed fMRI technology and found that language processing involved not just traditional language areas but also those involved in perceptual and motor simulations, with words related to specific sensory and motor experiences activating corresponding sensory and motor areas of the brain [11]. In some ways, these two studies are somewhat contradictory. Where LASS predicts that early in a word association task, language systems are predominantly used, focusing on linguistic and semantic processing of words, WFE predicts more readily accessible words will rely more heavily on non-language-based systems. It is possible that both language and non-language systems are engaged with varying degrees of dominance depending on the stage of the task and the nature of the word (abstract vs. concrete, high-frequency vs. low-frequency). Another explanation may be simply that the tasks explored in both studies (recall in Binder et al. versus generation in Simmons et al.) [7,11] were too different to compare. Furthermore, fMRI studies can suffer from methodological challenges, and have been criticized for low test-retest reliability for a range of cognitive tasks [49], making the ability to reproduce results all the more pressing.

Despite these challenges, we were inspired by the ability of fMRI to note precise activation times, as well as tentative evidence for LASS in a word association task gathered from a prior study [50] to pivot to developing an alternative, that while making no claims of being directly comparable to fMRI, might serve as an accessible and scalable approach to exploring conceptual processing without access to sophisticated equipment and large budgets. Through the development of a custom question in survey.js measuring the exact time keystrokes in a word association task, a word's relative frequency (*associative strength*) [19] and proximity to a cue word in vector space (*cosine similarity*) [51], we seek to explore the ways in which LASS and the WFE may help to provide a more nuanced and differentiated interpretation of the words generated in a timed, free word association task.

Although no studies could be directly found exploring semantic drift in the context of LASS, or for sub-minute timeframes, just as environmental changes, pandemics and even general conversations experience it, we expect to see a shift in the similarity of cue word to generated responses across a word association task as we process cue words from general to idiosyncratically situated associations [10,11,52]. The present study seeks to measure changes in meaning and the strength of association over time (semantic drift), specifically, the short time frames typical of word association tasks, and whether the simple addition of measuring *keystrokes* to such word association studies could represent a scalable method to explore cognition with finer granularity in word association studies.

## Hypotheses

Given the predictions of LASS that language systems will typically be invoked before situated simulation [10], and predictions of the WFE that the most frequent terms facilitate ease of recall [41], it is hypothesized that terms generated earlier in free word association tasks will exhibit higher levels of both semantic similarity and associative strength than those terms generated later in the task, due to the earlier activation of word associations, and availability of more frequent words. In other words, we expect that the associations with a cue word that people make early, will also tend to be words that often "socialize" with the cue word more generally. Our main research questions can be summarized as follows:

- RQ1: Can semantic drift be identified within a word association task?

- RQ2: Are the most frequent terms generated earlier?

This gives rise to the hypotheses that words generated earlier in the word association task, will have higher associative strength (greater overall representation in the corpus of terms generated) and semantic proximity to the cue word (H1) due to the prevalence of common words both in corpus-derived and human word associations [19]; that words generated later will be further apart (semantic similarity measures) from one another on average due to the greater heterogeneity (H2) among idiosyncratic, situated word associations [9,11], and that there will be fewer unique words generated earlier [41] in a word association tasks due to greater overall variability in latter associates (H3: more unique terms).

Finally, it is hypothesized that the effect sizes for H1-H3 will be stronger for concrete, common cue words, over abstract cue words (H4) due to different systems being activated for each type [7,10,11,52]. As corollaries to H1 and H3, we hypothesized that the most frequent words would also be semantically closer to the cue word and be generated earlier (H5a and b). All hypotheses (except for H5) were pre-registered at OSF.io (https://osf.io/k9xej/), and are summarized as follows (for RQ1, H1, H2, H4 and H5a, and RQ2, H3, H4 and H5b):

- H1: Terms generated in the 4th time quartile (Q4) of a word association task will be semantically more distant from the CUE word than those generated in the first quartile.

- H2: Terms generated in the 4th time quartile of a word association task will be semantically more distant from one another than those generated in the first quartile (variability).

- H3: Terms generated in the 4th time quartile of a word association task will exhibit greater diversity (more unique terms) than those generated in the first quartile.

- H4: Effect sizes of H1 to H3 will be stronger for concrete terms (tree, plastic, dog), and weaker for abstract terms (love, quality).

- H5 (Exploratory): Grouping terms by their overall relative frequency (associative strength), it is hypothesized that most common words in each word association task (cue) will both appear earlier and be semantically closer to the cue word.

   ○ H5a (corollary to H1): The most frequent terms will be semantically closer to the cue word on average.

   ○ H5b (corollary to H3): The most frequent terms will be generated earlier on average.

## Methods

### Survey

A survey was administered on survey.js, containing five-word associations (see S1 Appendix). The words (cues) were chosen to represent abstract and concrete terms on a scale of 1–5 (abstract to concrete). Abstract terms chosen were *love* (2.07) and *quality* (2.18), while *dog* (4.85), *tree* (5.00) and *plastic* (4.86) were chosen as concrete terms based on the concreteness ratings of 40,000 generally known English words [8].

The questions were presented in a timed word association with 25 seconds given to each cue word. We developed custom code for survey.js to enable every keystroke to be recorded with a timestamp, through which we were able to obtain precise measures of when a word was typed.

### Participants

The survey was advertised on the panel platform Prolific, targeting participants over 18 in the United Kingdom with all responses gathered October 10, 2023. 464 valid surveys were

submitted. Respondents were predominantly white (84.03%), female (62.4% female) with an average age of 40.8 years.

## Semantic similarity and associative strength

Cosine similarity scores were extracted from GloVe (Global Vectors for Word Representation) 840B 300d [53]. GloVe is a context free model, however, meaning for example, that the word *bank* as a place to deposit money and the land at either side of a river, would be represented as identical. Scores for each word to cue word pair were extracted from the GloVe data, for example [leaves, tree], and then a full distance matrix was generated for all words to all words to calculate average distance across quartiles. Semantic (or cosine) similarity is reported on a scale of -1 to 1.

To complement the static, global GloVe embeddings, we also generated embeddings using BERT (Embeddings from Language Models). Unlike GloVe, which provides static, context free word embeddings, BERT generates dynamic embeddings that are sensitive to the context in which a word appears (so *bank deposit* and *river bank* would be represented separately). This contextual awareness can be beneficial in analysing word associations that may depend on the specific context or usage of the words, however as BERT usually processes sentences and generates embeddings based on that context, the *cue* and *words* also needed to be given minimal context by placing them within sentences, for our purposes, of the form "in the context of [cue] one might think of [word]", an approach discussed in the literature [54–56]. Employing this sentence template provides the model with the context that, for example, the word *bark* in *tree_bark* likely refers to a tree rather than a dog.

A final set of measures was implement using WordNet. WordNet provides a structured representation of semantic relations between words, allowing for the analysis of associations based on taxonomic and conceptual relationships by organising them into synsets (sets of synonyms that share a common meaning) [57]. Despite being one of the older methods, WordNet has been found to match or even outperform modern processes employing embedding (such as BERT) in some contexts [58]. The Wu Palmer measure was chosen as it can help control for polysemy by controlling for the least common subsume (LCS), taking into account the different senses of cue words and their positions in the WordNet taxonomy. For example, if we compare the two senses of the word "bark"—"the outer layer of a tree" and "the sound a dog makes"—the LCS would be the more general concept of an "entity". However, if we compare "bark" (sound a dog makes) with "dog" (cue word provided to trigger a response), the LCS would be the more specific concept of "dog", resulting in a higher similarity score.

In addition to word embeddings and path distances, associative strength was calculated for each cue as the number of unique words per time quartile (Q1 1-7s, Q2 7-13s, Q3 13-19s, Q4 19–25 seconds), divided by the number of unique respondents [19], resulting in a score from 0 to 1.

## Hypothesis tests

Appropriate tests for group differences were employed. Due to the non-normal distribution of the dataset, group differences were tested using nonparametric methods including Wilcoxon Signed-Rank Test for within-participants comparisons for H1, and Mann-Whitney U and Z-tests for within group comparisons. Details are included in the results section for each hypothesis.

During the exploration of the data, two additional hypotheses were added as corollaries to the previous pre-registered hypotheses (H1 and H3), as if words generated later are both semantically more distant (H1) and the number of unique terms generated later greater (H3), it follows that another way to consider those questions would be to group words by their relative frequency (associative strength). As corollaries of H1 and H3, H5a and H5b were, therefore, added, as exploratory hypotheses.

### Ethics

Ethical approval was obtained from the Leeds Becket University Ethics Committee August 7, 2023. The exact text provided to participants can be found in the "Participation" section of the survey in the S1 Appendix.

## Results

### Sample

Table 1 provides a summary of participants sampled, demographics and the total and unique words generated for each cue in the word association task. The section then continues with the results for each hypothesis.

### Hypothesis 1

H1: Terms generated in the 4th time quartile of a word association task will be semantically more distant from the cue word than those generated in the first quartile.

- $H_0$: There is no significant difference in the semantic distance of terms in Q1 and Q4.

- $H_a$: There is a significant difference, with Q4 having greater semantic distance than Q1.

The hypothesis stated that terms generated in the fourth quartile (Q4; 19–25 seconds) would be more semantically distant from the cue words than those in the first quartile (Q1; 1–7 seconds). The range was chosen as all timestamps ranged from slightly over 1, to slightly less than 25.

As data were paired, a paired t-test would be appropriate for normally distributed data, or a Wilcoxon Signed-Rank test for non-parametric data. To determine whether a paired t-test would be suitable, the data were paired by cue word and id if they had values in both time quartiles 1 and 4 (Q1 and Q), and valid values for similarity (SemSim, Wu Palmer). The paired samples (Q1 and Q4 pairs) were then evaluated for normality using Shapiro-Wilk tests.

The findings presented in Table 2 support rejection of the null hypothesis for H1, demonstrating small to moderate, significant reductions in semantic similarity across almost all cue words with the exception of *tree*, which despite a significant and moderate negative trend for

**Table 1. Descriptive statistics.**

| Associations generated | Total | Unique | Total Participants | 464 |
|---|---|---|---|---|
| ■ ALL | 10219 | 1569 | Gender—Female (%) | 62.40 |
| ■ tree | 2251 | 318 | Gender—Male (%) | 37.30 |
| ■ dog | 2492 | 380 | Gender–Not stated (%) | .29 |
| ■ quality | 1550 | 423 | Ethnicity—White (%) | 84.03 |
| ■ plastic | 1810 | 373 | Ethnicity—Asian (%) | 5.50 |
| ■ love | 2116 | 393 | Ethnicity—Mixed (%) | 5.24 |
| Average Words per Person | 22.02 | | Ethnicity—Black (%) | 3.83 |
| Average Age | 40.80 | | Ethnicity—Other (%) | 1.35 |
| ■ Age Range 18–24 (%) | 10.34 | | Ethnicity–Not stated (%) | .06 |
| ■ Age Range 25–34 (%) | 23.28 | | | |
| ■ Age Range 35–44 (%) | 21.98 | | | |
| ■ Age Range 45–54 (%) | 16.81 | | | |
| ■ Age Range 55–64 (%) | 18.32 | | | |
| ■ Age Range 65+ (%) | 1.08 | | | |

**Table 2. Hypothesis 1 with Wilcoxon signed-rank test and paired t-tests across all NLP methods.**

| cue | Q1 mean (SD) | Q4 mean (SD) | Q1 med | Q4 med | S.-Wilk | NLP | Test Stat | Stat | Sig. | Effect | N | Dir. |
|---|---|---|---|---|---|---|---|---|---|---|---|---|
| tree | .48 (.07) | .41 (.11) | .50 | .41 | .02 | GloVe | WSRT | 3627.50 | .00 | .50 | 184 | - |
| | .79 (.04) | .80 (.05) | .78 | .80 | .02 | BERT | WSRT | 6493.50 | .01 | .21 | 184 | + |
| | .31 (.13) | .32 (.19) | .29 | .27 | .00 | WordNet | WSRT | 8388.00 | .87 | 618.37 | 184 | ns |
| dog | .53 (.15) | .40 (.13) | .52 | .38 | .69 | GloVe | t-test | 9.91 | .00 | .67 | 220 | - |
| | .82 (.05) | .77 (.06) | .82 | .77 | .82 | BERT | t-test | 8.79 | .00 | .59 | 221 | - |
| | .52 (.24) | .33 (.24) | .50 | .23 | .00 | WordNet | WSRT | 4923.00 | .00 | 332.67 | 219 | - |
| quality | .44 (.12) | .38 (.13) | .45 | .39 | .91 | GloVe | t-test | 4.26 | .00 | .35 | 152 | - |
| | .78 (.04) | .76 (.06) | .78 | .77 | .77 | BERT | t-test | 2.26 | .03 | .18 | 152 | - |
| | .43 (.19) | .38 (.18) | .40 | .30 | .14 | WordNet | t-test | 2.67 | .01 | .22 | 148 | - |
| plastic | .37 (.11) | .34 (.12) | .38 | .32 | .22 | GloVe | t-test | 3.22 | .00 | .23 | 190 | - |
| | .78 (.05) | .77 (.05) | .78 | .77 | .93 | BERT | t-test | 2.79 | .01 | .20 | 190 | - |
| | .30 (0.12) | .27 (0.10) | .28 | .25 | .00 | WordNet | WSRT | 6190.00 | .01 | 452.66 | 187 | - |
| love | .48 (.10) | .42 (.12) | .48 | .44 | .01 | GloVe | WSRT | 6581.00 | .00 | .32 | 204 | - |
| | .85 (.05) | .82 (.06) | .85 | .82 | .05 | BERT | WSRT | 5666.00 | .00 | .40 | 204 | - |
| | .41 (0.22) | .37 (0.22) | .33 | .29 | .34 | WordNet | t-test | 1.80 | .07 | 0.13 | 202 | ns |

GloVe, exhibited a tiny yet significant increase in similarity, and no significant difference for Wu Palmer (WordNet). The cue *love* was not significant for WordNet either, though exhibited moderate negative trends for GloVe and BERT. Fig 1 provides paired-sample density plots for semantic similarity over time, for each cue and NLP method.

## Hypothesis 2

Terms generated in the 4th time quartile of a word association task will be semantically more distant from one another than those generated in the first quartile (variability).

- $H_0$: There is no significant difference in the inter-word variability of terms between Q1 and Q4.

- $H_a$: There is a significant difference, with Q4 having greater inter-word variability than Q1.

For each cue word, similarity matrices for Q1 and Q4, which included semantic similarity scores for all word pairs were created. The semantic distance as 1 minus the similarity score was then calculated, along with the average semantic distance and standard deviation for each cue word in both quartiles. The Mann-Whitney U test was used to compare distributions of semantic distances within Q1 and Q4 for each cue word and Cliff's Delta calculated for each cue word, comparing semantic distances of word pairs in Q1 against those in Q4.

The analysis of the similarity matrices for the first and fourth quartiles (Q1 and Q4) for all cues and words yielded the average inter-term semantic distances presented in Table 3.

These values suggest that, with the exception of the cue *tree*, words generated to most cues in the fourth quartile (Q4) were not consistently more distant from the cue than those in the first quartile (Q1). The density plots in Fig 2 illustrate the almost identical distributions from Q1 to Q4.

We do not find, therefore, compelling evidence to reject the null hypothesis that words generated in the fourth quartile are not semantically more distant from one another than those generated in the first quartile.

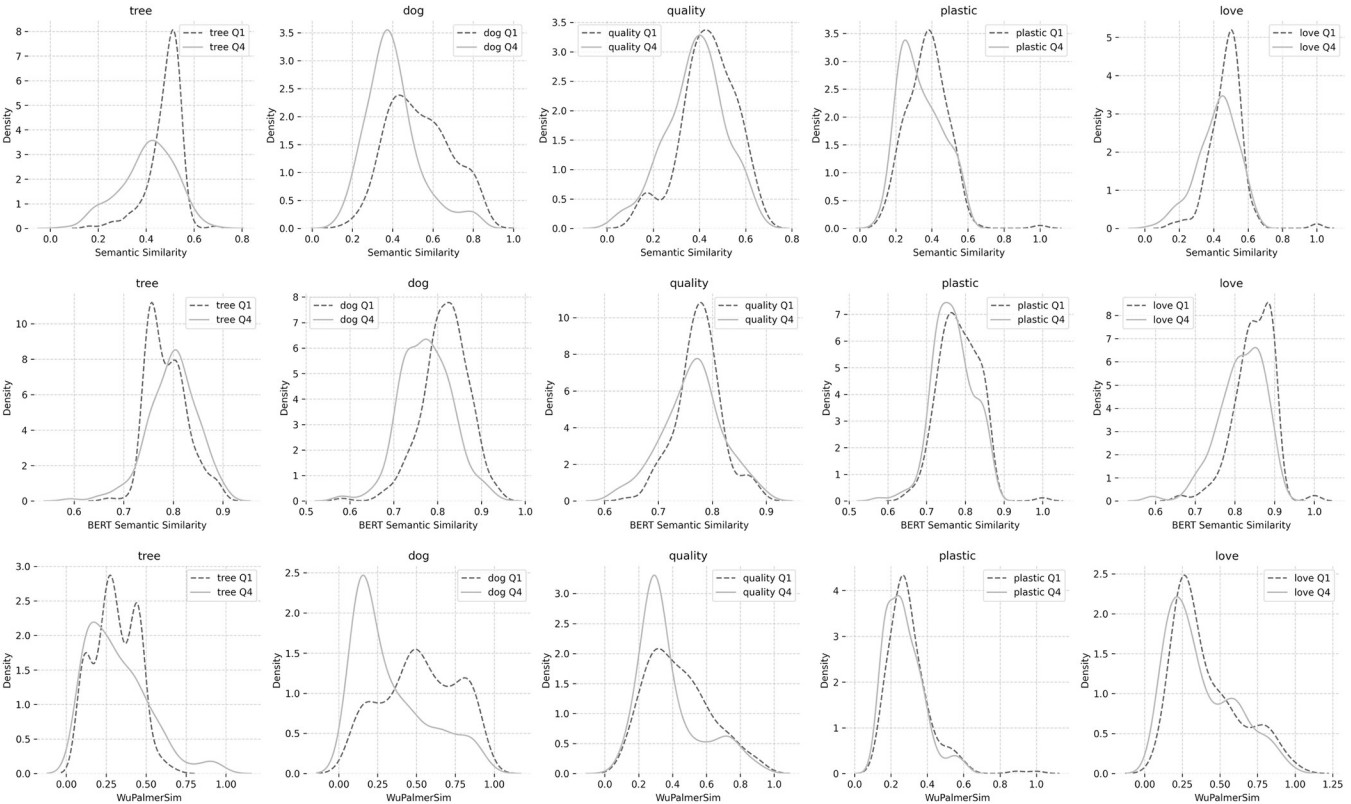

**Fig 1. Density plots for semantic similarity scores across quartiles, cues and NLP methods.**

## Hypothesis 3

H3: Terms generated in the 4th time quartile of the word association task will exhibit greater diversity (more unique terms) than those generated in the first quartile.

**Table 3. Hypothesis 2 with Mann-Whitney U test.**

| Cue Word | Q1 mean (SD) | tQ4 mean (SD) | NLP | U-Statistic | p-Value | Cliff's Delta |
|---|---|---|---|---|---|---|
| tree | .23 (.13) | .21 (.13) | GloVe | 29609826 | p < .001 | .08 |
| | .92 (.05) | .90 (.08) | BERT | 30200326.5 | p < .001 | .10 |
| | .15 (.20) | .13 (.18) | WordNet | 29964010.5 | P < .001 | .09 |
| dog | .24 (.13) | .22 (.13) | GloVe | 73726325 | p < .001 | .05 |
| | .92 (.06) | .92 (.07) | BERT | 69981000.5 | p < .001 | -.05 |
| | .10 (.20) | .10 (.19) | WordNet | 72612966.5 | p < .01 | -.02 |
| quality | .26 (.13) | .24 (.13) | GloVe | 67192996 | p < .001 | .11 |
| | .93 (.05) | .92 (.06) | BERT | 62427898.5 | p = .73 | .00 |
| | .08 (.14) | .08 (.14) | WordNet | 62100803.5 | P = .70 | .00 |
| plastic | .22 (.12) | .22 (.12) | GloVe | 59817487 | p = .76 | .00 |
| | .91 (.06) | .92 (.06) | BERT | 60094395 | p = 16 | -.01 |
| | .09 (.17) | .10 (.17) | WordNet | 60330884.5 | P = 28 | -.01 |
| love | .27 (.14) | .25 (.14) | GloVe | 60806690 | p < .001 | .09 |
| | .94 (.04) | .93 (.05) | BERT | 54236010.5 | p < .005 | -.03 |
| | .13 (.19) | .12 (.19) | WordNet | 58516777.5 | p < .001 | .05 |

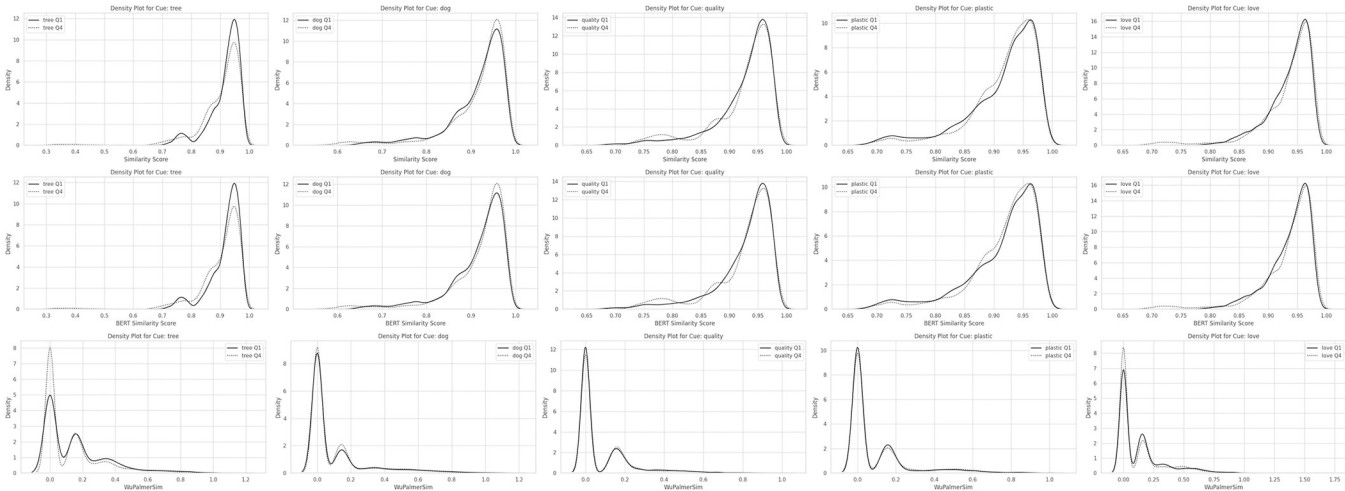

**Fig 2. Density plots for term x term semantic similarity scores across quartiles.**

- $H_0$: There is no significant difference in the diversity of terms in Q1 and Q4.

- $H_a$: There is a significant difference, with Q4 having more unique terms than Q1.

Due to the differences in absolute words generated across each quartile, a relative measure of diversity was chosen (associative strength, as a proportion of unique terms in the total for each). Given that proportions (relative uniqueness ratios) were to be compared, a Z-test for two proportions was chosen to determine the statistical uniqueness of the two populations (Q1 and Q4). All cues met the sample size and proportion conditions for the Z-test in both Q1 and Q4, indicating that the assumptions for the test were satisfied. Table 4 provides a summary of the results of the Z tests and the relative frequencies of unique terms for each quartile and cue.

A Cohen's h of -.5 to -.6 it suggests a medium to large effect size. Specifically, the results suggest that the proportion of unique terms in the first group (Q1) is significantly lower than in the second group (Q4), and the magnitude of this difference is substantial. In practical terms, this can mean that the change in proportions between the two quartiles is not only statistically significant but also of a meaningful size, indicating a considerable shift in the data characteristics from Q1 to Q4.

To test whether these findings remained robust on paired samples, we also ran a Wilcoxon Signed-Rank Test for each cue and paired Q1 to Q4, sample. Table 5 includes the mean and standard deviation (SD) for the first (Q1) and fourth (Q4) time quartiles, along with the test statistic, p-value, and effect size from the Wilcoxon Signed-Rank Test for each cue. The data indicates significant differences in associative strength between Q1 and Q4 across all cues, with the more frequently cited words (WFE) occurring earlier in the task than the less frequently occurring words.

**Table 4. Relative uniqueness by quartile with Z tests.**

| Cue | Z | p-value | Cohen's h | %unique Q1 | %unique Q4 |
|---|---|---|---|---|---|
| tree | -8.53 | < .001 | -.53 | 14.82 | 37.81 |
| dog | -10.29 | < .001 | -.60 | 16.34 | 43.32 |
| quality | -7.69 | < .001 | -.60 | 30.26 | 59.83 |
| plastic | -8.13 | < .001 | -.58 | 24.79 | 52.50 |
| love | -9.64 | < .001 | -.62 | 18.80 | 47.43 |

**Table 5. Wilcoxon Signed-Rank Test for associative strength (cue strength).**

| cue | Q1 mean (SD) | Q4 mean (SD) | Q1 median | Q4 median | Shapiro-Wilk | Test Stat | p | Effect Size | N |
|---|---|---|---|---|---|---|---|---|---|
| tree | .12 (.07) | .02 (.02) | .12 | .01 | < .05 | 263 | < .001 | .02 | 184 |
| dog | .05 (.04) | .01 (.01) | .05 | .01 | < .05 | 1268 | < .001 | .05 | 221 |
| quality | .04 (.04) | .02 (.02) | .02 | .01 | < .05 | 3124 | < .001 | .27 | 152 |
| plastic | .03 (.03) | .01 (.01) | .02 | .01 | < .05 | 3563 | < .001 | .20 | 190 |
| love | .06 (.06) | .01 (.01) | .04 | .01 | < .05 | 2413 | < .001 | .12 | 204 |

To obtain a measure of diversity within each group we also calculated the Shannon Diversity Index [59] as it considers both the richness (the number of different words) and the evenness (the distribution or abundance of each word) within the dataset. The results presented in Table 6 provide further evidence for increases in diversity over time (higher H values indicate greater diversity). Given the findings across these tests, the null hypothesis for H3 can be rejected.

## Hypothesis 4

H4: Effect sizes of H1 to H3 will be stronger for concrete terms, and weaker for abstract terms. Concrete terms were defined as [tree, dog, and plastic], while abstract terms were defined as [love and quality].

- $H_0$: H1 to H3 are not significantly stronger for concrete terms, and weaker for abstract terms.

- $H_a$: H1 to H3 are significantly stronger for concrete terms, and weaker for abstract terms.

    Considering the results for H1 to H3:

- Considering the results for H1 (Table 2), the effect sizes did not show any consistently stronger effects for concrete terms, though *dog* did have by a large margin the highest effect size for drops in semantic similarity from tQ1 to tQ4.

- For H2 (Table 3 and Fig 2) the results were similar. No meaningful differences could be found for concrete versus abstract words in terms for variance in tQ4 versus tQ1.

- For H3 (Tables 4–6, and Fig 3) effect sizes were generally similar across the cues.

Overall, the null hypothesis for H4 could not be rejected, and there were no consistent differences across abstract and concrete terms for H1 to H3.

## Hypothesis 5 (exploratory)

H5a: Grouping terms by their frequency, it was hypothesized that the most common words in each word association task (cue) will be semantically closer to the cue word.

**Table 6. Shannon Diversity Index (H) for each community (cue x timeQuartile) and species (word).**

| Cue | Q1 SDI (H) | Q4 SDI (H) |
|---|---|---|
| tree | 4.37 | 6.56 |
| dog | 5.42 | 6.74 |
| quality | 6.02 | 6.69 |
| plastic | 5.94 | 6.76 |
| love | 5.51 | 6.78 |

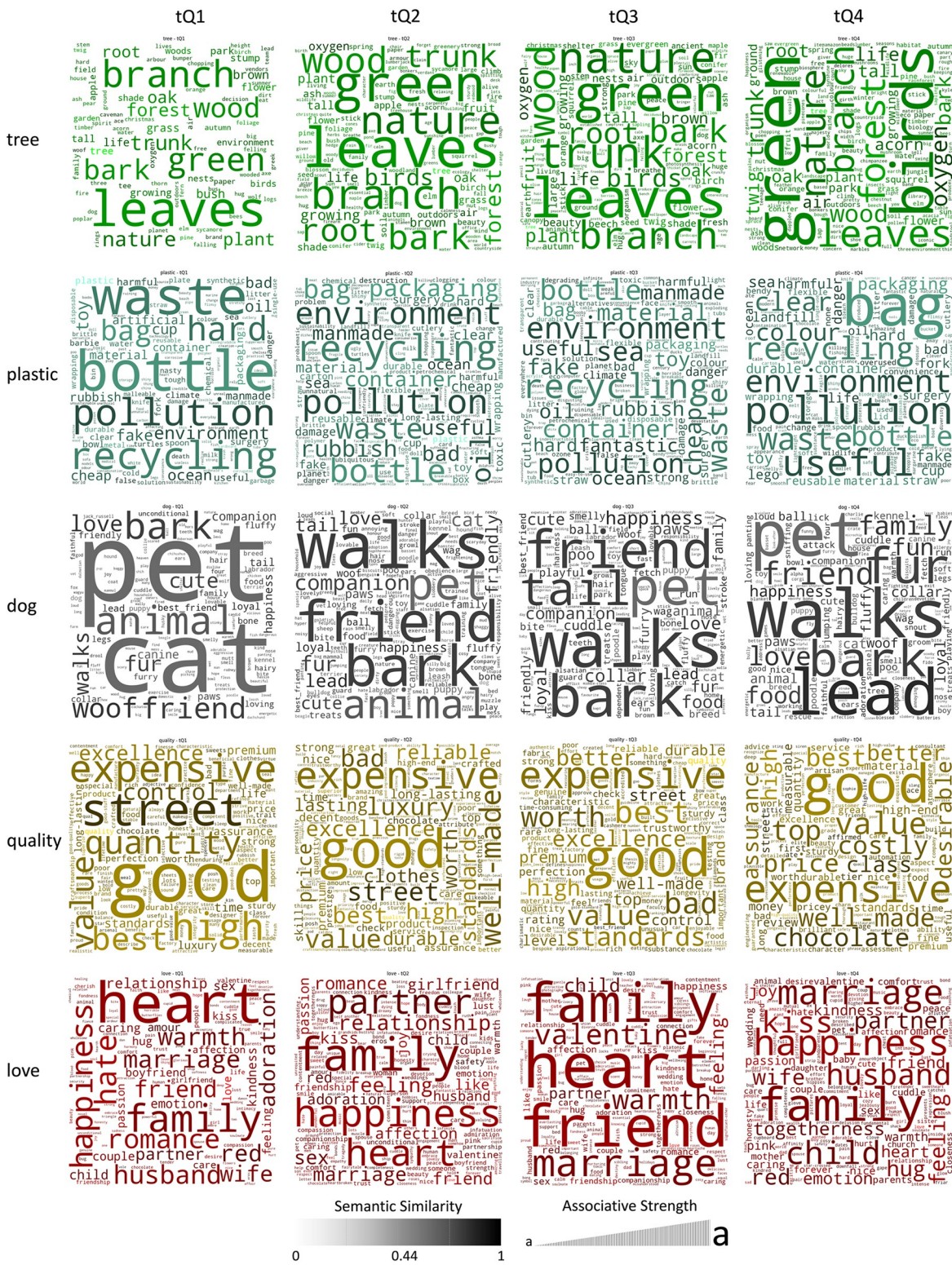

**Fig 3. Word clouds by associative strength and semantic similarity (Q1 to Q4) for GloVe.**

**Table 7.**

| cue | Q1 mean (SD) | Q4 mean (SD) | Q1 med | Q4 med | S.-Wilk | NLP | Test Stat | Stat | Sig. | Effect | N | Dir. |
|---|---|---|---|---|---|---|---|---|---|---|---|---|
| tree | .50 (.02) | .34 (.09) | .50 | .34 | .79 | GloVe | t-test | 25.28 | < .001 | 2.44 | 216 | - |
| | .75 (.00) | .79 (.05) | .75 | .79 | .056 | BERT | t-test | -9.926 | < .001 | -.97 | 216 | + |
| | .27 (.10) | .29 (.18) | .27 | .25 | .000 | WordNet | WSRT | 1118.500 | .811 | .486 | 214 | ns |
| dog | .62 (.14) | .34 (.10) | .62 | .33 | .24 | GloVe | t-test | 25.35 | < .001 | 2.29 | 223 | - |
| | .83 (.05) | .75 (.06) | .83 | .75 | .442 | BERT | t-test | 16.576 | < .001 | 1.48 | 223 | - |
| | .62 (.23) | .34 (.22) | .66 | .28 | .000 | WordNet | WSRT | 2694.500 | .000 | .112 | 219 | - |
| quality | .51 (.10) | .30 (.11) | .52 | .31 | < .05 | GloVe | WSRT | 213 | < .001 | .02 | 142 | - |
| | .78 (.02) | .75 (.06) | .78 | .75 | .062 | BERT | t-test | 4.932 | < .001 | .58 | 146 | - |
| | .51 (.19) | .33 (.13) | .51 | .29 | .001 | WordNet | WSRT | 1085.000 | .000 | .112 | 139 | - |
| plastic | .40 (.11) | .28 (.09) | .43 | .28 | < .05 | GloVe | WSRT | 1303 | < .001 | .11 | 157 | - |
| | .79 (.05) | .76 (.06) | .79 | .76 | .420 | BERT | t-test | 5.910 | < .001 | .59 | 157 | - |
| | .27 (.12) | .26 (.09) | .24 | .25 | .013 | WordNet | WSRT | 5882.000 | .988 | .487 | 155 | ns |
| love | .49 (.06) | .35 (.12) | .50 | .35 | .90 | GloVe | t-test | 13.27 | < .001 | 1.39 | 179 | - |
| | .86 (.04) | .77 (.07) | .87 | .78 | .470 | BERT | t-test | 15.121 | < .001 | 1.63 | 179 | - |
| | .38 (.18) | .28 (.14) | .29 | .24 | .004 | WordNet | WSRT | 3918.500 | .000 | .246 | 178 | - |

- $H_0$: There is no significant difference in term frequency and semantically proximity.

- $H_a$: The most frequent terms will be semantically closer to the cue word on average.

  H5b: Grouping terms by their frequency, it was hypothesized that the most common words in each word association task (cue) will appear earlier.

- $H_0$: There is no significant difference in term frequency and generation time.

- $H_a$: The most frequent terms will be generated earlier on average.

  To conduct these tests, words were organized into approximate quartiles of frequency for each word association test (cue), and the average initial timestamps and semantic similarity values calculated.

  H5a posited that the most frequent terms would be semantically closer to the cue word on average. Table 7 provides the results for the Wilcoxon Signed-Rank and t-tests and relevant summary statistics. Some very large sizes were observed, indicating highly relevant and practically significant results for specific cues, while overall support was found for the alternate hypothesis, despite some reversals (*tree* for BERT, and *plastic* and *love* for WordNet).

  The null hypothesis ($H_0$) could be rejected for this hypothesis, with very strong effect sizes for three of the cues (Effect Size, Table 7), and small yet significant effects sizes for two of the five cues, supporting the alternate hypothesis ($H_a$) for all cue words.

  H5b posited that the most frequent terms would be generated earlier on average. The results were highly significant with medium effect sizes, suggesting that the differences are both significant and meaningful across all cues and for most of the NLP methods used, though some effects were reversed for BERT and WordNet. Overall, however, the null hypothesis ($H_0$) could, therefore, be rejected with at least two out of three tests showing the predicted effects for all cue words. The results are provided in Table 8.

  Finally, to provide a visualisation of the terms generated, Fig 3 organises words by cue, and quartile (for GloVe). Word clouds for BERT and Wordnet Wu Palmer can be generated via the codebooks. Similarity within the cue is represented by shading, with lighter terms more distant from the cue, and darker words closer. Size represents the overall associative strength

**Table 8. (H5b) Generation time by word frequency (Wilcoxon Signed-Rank Test).**

| cue | Q1 mean (SD) | Q4 mean (SD) | Q1 median | Q4 median | Shapiro-Wilk | Test | Test Stat | p | Effect Size | N |
|---|---|---|---|---|---|---|---|---|---|---|
| tree | 8.57 (4.69) | 15.11 (4.60) | 7.75 | 15.47 | < .05 | WSRT | 1792 | < .001 | .08 | 216 |
| dog | 8.12 (4.60) | 13.64 (4.90) | 7.46 | 13.70 | .16 | t-test | -12.16 | < .001 | -1.16 | 223 |
| quality | 9.47 (5.77) | 13.46 (5.77) | 8.55 | 13.60 | < .05 | WSRT | 2727 | < .001 | .25 | 146 |
| plastic | 9.74 (5.44) | 13.47 (5.68) | 8.90 | 14.19 | < .05 | WSRT | 3326 | < .001 | .27 | 157 |
| love | 8.98 (5.15) | 13.93 (5.11) | 8.94 | 14.10 | < .05 | WSRT | 2660 | < .001 | .17 | 179 |

of the term within the group (unique words divided by unique respondents). The relative frequency of terms can be tracked within the quartiles, for example, for GloVe (Fig 3) *bottle*, is frequently cited in tQ1 for *plastic*, but diminished in the remaining quartiles. Similarly, *birds* was cited by few respondents in tQ1, while it represents one of the largest terms in tQ4, offering potential evidence for a shift in language to situated simulation, by way of the frequency and semantic distances of terms over time.

## Discussion

Before discussing the results, it is important to provide some context around effect sizes, which we have provided in addition to the unique measure for each test of group differences, both for ease of comparison, and future (meta) studies. As a rule, it must be noted when discussing effect sizes, that small to moderate effects are common in psychological research, and can be very meaningful, especially in large-scale studies [60]. Furthermore, a study investigating differences among the magnitude of effect sizes in pre-registered and unregistered studies found reported effects were generally substantially lower in pre-registered studies (r = .16) than unregistered studies (r = .36), cited as evidence of biases and potentially questionable research practices [60]. Although we were unable to reject H2 (stronger effect sizes for abstract versus concrete terms), the moderate and significant effect sizes we obtained for H1, H3 and H5a and b, may be regarded as promising.

Our hypotheses were designed to explore two main research questions: 1. Can semantic drift be identified over time within a word association task, and 2. Are frequent terms also generated earlier? To test for the presence of semantic drift, Hypothesis 1 stated that terms generated in the 4th time quartile of a word association task would be semantically more distant from the cue word than those generated in the first quartile. Across all cue words, H1 held for GloVe, and most for BERT and WordNet. Terms generated in Q4 were indeed, on average, semantically more distant from the cue word than words generated in Q1, not only consistently suggesting the presence of semantic drift across the word association task when using a fixed, global embedding model such as GloVe, but also largely supported by the dynamic contextual embeddings provided by BERT, and the Wu Palmer similarity formula of WordNet. Of interest, the cue *dog*, elicited the greatest effect sizes and differences, most notably for WordNet (a difference of almost 0.2, four times higher than the next highest for *quality* at .05), suggesting that some cues may lend themselves to observing stronger effects for LASS than others. This may be due in part to the multimodal nature of the concept of dogs themselves, in that they evoke visual, auditory, tactile and motor simulations [9], are both concrete conceptually [5,6], yet evoke rich personal experiences, and may be strongly emotionally grounded [61].

The t-tests and Wilcoxon signed-rank tests (Table 2) confirmed significant evidence for semantic drift from the first to fourth quartile. These results are further supported by the distributions in Fig 1. Semantic drift was reversed for the cue *tree* for BERT, and no significant differences were found for *plastic* or *quality* in terms of the path distance of WordNet. That

some differences might exist across models is not unexpected considering GloVe and BERT have fundamentally different architectures and differences in the text corpora used to train them both. Different ways too of providing minimal cues and responses for the BERT model might yield varying results, as the comparison is made between the whole sentences. Regarding WordNet, whereas GloVe and BERT represent semantics in a dense vector space, WordNet uses a graph-based representation with synsets. This difference in representation could also lead to variations in capturing semantic drift, ultimately reinforcing the importance the choice of model and the underlying assumptions they make will have in terms of influencing the relevance and value of the findings of a given research question.

As a corollary to H1, we reframed the hypothesis in terms of the word frequency effect (WFE), predicting that the most frequent terms would be semantically more similarity to the cue words. We found strong effects for several of the cue words, most notably *dog*, *tree* and *love* (Table 7). Overall, the most frequent words were semantically closer to the cue word than the least frequent terms as predicted. Our second corollary posited that the most frequently generated terms, would be generated earlier (Table 8). This too was supported by the results of the analysis, with the most frequent terms generated around 5–6 second earlier on average that the least frequent terms.

These hypotheses combined provide evidence and offer promising answers to our core research questions, whereby words generated earlier are semantically closer to the cue word (as predicted by LASS), and that the most frequent words (WFE) are also generated earlier and are semantically closer to the cue in a free word association task. That said, not all cues generated the predicted level of distance over time equally, suggesting some words may be more powerfully present across a range of representational types (from language to situated simulation), *dog* being perhaps the most consistently prominent. This might represent a near perfect example of a cue that might be expected to generate a more distinct range of terms, given both language and personal representation should exist in abundance, and represents an area for future research with a larger set and range of cues and cue types.

Given support found for the presence of semantic drift over time, we also hypothesized that if words were semantically less related to the cue word, they would also be sparser, and more distant from one another (H2). To explore this, we generated a full *term x term* matrix for GloVe, BERT, and WordNet synsets, for each cue and quartile. The resulting sets comprised the semantic distance measures for 1569 unique pairs, from which the values comparing the unique term matrices for Q1, and Q4 in Table 3 and Fig 2 were derived. For example, Q1 for tree included 4005 [n(n-1)/2] unique cosine similarity values and word pairs when excluding *self x self*-comparisons and duplicates. Largely significant, yet very small to negligible effects were found (Table 3, Cliff's Delta). Fig 2 illustrates very closely overlapping distributions. While the terms generated exhibited slightly greater sparsity, further adding to the growing evidence for semantic drift for GloVe, results were less marked for BERT and WordNet, and so should be interpreted as very tentative evidence at best. We failed to reject the null hypotheses despite some support, simply because the effect sizes were so small when present, that they would be of little if any practical importance. There may be variations that are derived from the concreteness and familiarity of the terms [8], or by the system switching from language associations to simulation predicted by LASS [10,11]. Alternately, the variability for the cues chosen, may also simply, not be large. As differences were present across some cues, future research might again, consider a larger range of cue words as a foundation.

Another expected corollary of semantic drift, Hypothesis 3 aimed to explore the prediction that associative strength, via the *Word Frequency Effect* (WFE), would see relatively fewer unique words generated earlier on, as more frequent words were expected to dominate, while later in the association task, potentially through situated simulations (LASS), we predicted a

relative increase in unique terms. This was robustly supported with medium effect sizes, and the proportion of unique words 20% to 30% lower in the first time-quartile as compared to the fourth across all cues (Tables 4–6). We also conducted a paired-samples analysis of the cue strength (associative strength) for Q1 and Q4, finding small yet highly significant results for each cue. Finally, we calculated the *Shannon Diversity Index* (SDI) for terms in the first- and fourth-time quartiles, further supporting the alternate hypothesis, that if associations invoke simulation over language associations, the terms generated will be more idiosyncratic and diverse [9–11]. As expected, terms generated in the fourth time quartile exhibited significantly greater diversity than those generated in the first-time quartile.

We could find no compelling evidence for differences in concrete versus abstract cues (H4), suggested by prior research [7,8]. While differences did exist across cues (e.g. Fig 1, notably for *dog*, and somewhat less pronounced for *quality* and *plastic*), the differences did not neatly fit a concrete versus abstract dichotomy. As we only tested 5 cue words, and our primary aim was to detect semantic drift, our inability to reject the null hypothesis for H4 may be a matter of requiring a dedicated study across a more diverse range of pre-rated terms as provided by prior research [8]. This is worthy of consideration both in terms of validating the current results with a more diverse set of terms, potentially across a larger sample, and in terms of exploring concreteness in the context of LASS and WFE.

Overall, the results for Hypotheses 1, 5a and 5b provided promising evidence that semantic drift can be identified within a word association task, while Hypothesis 3 and 5b, offered support for the idea that the most frequent terms are generally generated earlier. Predictions resulting from an interpretation of both LASS and WFE were, therefore, given further support through our findings, contributing to LASS and dual processing systems more generally, and to the word frequency effect (WFE). Our findings indicate that the LASS theory may profit from greater exploration of the semantic distance of words to cues, generated by the two interacting systems, with distance (or similarity) a potential, scalable indicator of the invocation of language versus simulation systems. Although the WFE has been explored extensively in many contexts [41], no research could be found considering the timing of a word relative to its frequency within a word association task. While limited prior research has considered word generation order as an indicator of the activation of different systems of cognition, by adding a means to explore the WFE in terms beyond frequency alone, we provide a way to explore dual processing theories of language, such as LASS, whilst adding a new dimension to the WFE itself, time. In short, we find combining LASS (or similar dual processing theories) and the WFE a novel and promising approach to obtaining greater insights into both effects.

Our research also has practical relevance. As a matter of convenience, the simple addition of code to enable the exact measurement of term generation by way of keystrokes, enables the subtle differences in perception and opinion to be explored with a fine level of granularity. To further standardise and simplify analysis, there may be even simpler methods, such as using dedicated text fields in surveys where a timestamp is added when the user progresses to the next field, or the question times out, though this would not be as accurate as the *keystrokes* methods applied here, it could be sufficiently detailed to scale further. For practitioners, given the results of this study, being able to easily explore when terms were generated could offer a powerful means of exploring the diffusion of awareness around topics (e.g. the salience of the term *microplastics* for ecology), or associations with a brand, or destination [50]. Indeed, both the speed and frequency with which terms are generated represent a promising measure of the effectiveness of marketing and awareness campaigns more generally. The transition of brands to everyday nouns is an excellent example of how powerful the word frequency effect, combined with semantic proximity, can be. Whether we are *Ubering* to an important meeting, *Photoshopping* an image, or *Googling* how to *Xerox* a large number of copies of our research

papers, words such as *Fiberglass*, *Frisbee*, *Hoover*, and *Styrofoam* have become synonymous with the product category they represent. The semantic proximity of these terms to the concept or object they represent, frequency of use and speed of recall, are all examples of the confluence of LASS and WFE.

The present study provides support for the idea that the relationships words have to one another changes, whether due to pandemics [36], conceptual drift through history [24], the space of a single conversation [39], or as presented here, within a condensed, 25-second, word-association task. Furthermore, it illustrates the importance of understanding context and meaning when interpreting word association data, and that timing, as in so many aspects of life, matters.

## Limitations and future work

As with any study, our research too has limitations. Being primarily an initial exploratory study, to reduce the cognitive load for participants, we only included a small set of word associations, totalling 5 cues per person. Due to the differences across cues in terms of effect sizes, with some cues, most notably, *dog*, exhibiting consistently stronger effect sizes across all models, we might expect certain words to be more likely to induce simulations than others. The potential for certain words to be more visceral, or memorable, might have implications for a range of disciplines, from counselling and coaching to marketing. We intend therefore, to explore the hypotheses of this paper further, with a much larger set of cues, and cue types (for example concrete, abstract, living, non-living, etc).

Another limitation is that we did not seek a representative sample of respondents, recruiting freely from the Prolific platform. As such, it was not possible to robustly explore whether there may be variations by sex, age, ethnicity, or any other demographics. This too represents an opportunity for future research designs.

Finally, by recruiting from an online panel service, the results cannot be generalised to other populations.

## Conclusion

Our research provides initial evidence for the presence of semantic drift in timed word association tasks, and that both LASS and WFE may offer partial explanations for how and why and when certain words are generated in word association tasks, and that the magnitude of the effects may be linked to the how the cues themselves are represented within language and experience. Cues such as *dog*, being both a common concept within language with well-established related concepts (such as cat animal, etc.), and for many, a highly emotional character, might lead to particularly significant effects. In summary, however, given the limitations stated, the first words people associated with the cue words were indeed, generally more closely related (LASS) and less diverse (WFE), while those generated later were both relatively more diverse (Shannon Diversity Index) and semantically more distant.

The study represents a simple, scalable, and cost-effective approach to exploring new levels of meaning within word association tasks, and thereby provides a promising approach for researchers and practitioners engaging in perception studies that make use of timed, free-word-associations. It also offers a means of measuring the diffusion of influence over time, as repeated measures may be taken at different times to observe the speed and frequency with which words drift into prominence, whether within an individual mind, a group, or a society. We believe the addition of timed *keystrokes* to word association studies represents a valuable approach to exploring semantic drift in a controlled, targeted, and scalable way, and may just

help us to better understand how presentation and representation conspire to generate our individual and shared realities.

## Supporting information

**S1 Appendix.**
(DOCX)

## Acknowledgments

Special thanks to Maxime MacNiven, who cross-checked the analysis and provided code for the survey.js custom question that was used for acquisition of timed word association data.

## Author Contributions

**Conceptualization:** Sean MacNiven.

**Data curation:** Sean MacNiven.

**Formal analysis:** Sean MacNiven.

**Funding acquisition:** Sean MacNiven, Ralph Tench.

**Investigation:** Sean MacNiven.

**Methodology:** Sean MacNiven.

**Project administration:** Sean MacNiven.

**Resources:** Sean MacNiven.

**Software:** Sean MacNiven.

**Supervision:** Sean MacNiven.

**Validation:** Sean MacNiven, Ralph Tench.

**Visualization:** Sean MacNiven.

**Writing – original draft:** Sean MacNiven.

**Writing – review & editing:** Sean MacNiven, Ralph Tench.

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
