## [Decision Letter · Decision Letter 0]

25 Mar 2024

PONE-D-24-02236Keystrokes: A practical exploration of semantic drift in timed word association tasksPLOS ONE

Dear Dr. MacNiven,

Thank you for submitting your manuscript to PLOS ONE. After careful consideration, we feel that it has merit but does not fully meet PLOS ONE’s publication criteria as it currently stands. Therefore, we invite you to submit a revised version of the manuscript that addresses the points raised during the review process.

We look forward to receiving your revised manuscript.

Kind regards,

Laura Morett

Academic Editor

PLOS ONE

Journal Requirements:

2. Please note that in order to use the direct billing option the corresponding author must be affiliated with the chosen institute. Please either amend your manuscript to change the affiliation or corresponding author, or email us at plosone@plos.org with a request to remove this option.

Reviewers' comments:

Reviewer's Responses to Questions

**Comments to the Author**

1. Is the manuscript technically sound, and do the data support the conclusions?

Reviewer #1: Yes

Reviewer #2: Yes

2. Has the statistical analysis been performed appropriately and rigorously? 

Reviewer #1: Yes

Reviewer #2: Yes

3. Have the authors made all data underlying the findings in their manuscript fully available?

Reviewer #1: Yes

Reviewer #2: Yes

4. Is the manuscript presented in an intelligible fashion and written in standard English?

Reviewer #1: No

Reviewer #2: Yes

5. Review Comments to the Author

Reviewer #1: The paper investigates semantic drift through the lenses of Language and Situated Simulation (LASS) and the Word Frequency Effect (WFE) within a timed word association task. The study hypothesizes that terms generated later in the task will be semantically more distant from the cue word, indicating semantic drift, and that more frequent terms are generated earlier. Utilizing GloVe 300B word embeddings and analyzing 1569 unique term pairs for five cue words, the study finds significant evidence of semantic drift, supporting the hypothesis of semantic distance increasing over time within participants. Additionally, it observes a decrease in the diversity of terms generated early in the task, aligning with the expectation that frequently used words dominate early stages. The findings contribute to the understanding of LASS, WFE, and semantic drift, offering a scalable method to explore changes in semantic relationships. However, the paper suffers from several significant shortcomings that limit its impact and utility in the broader research context.

Firstly, the writing style is notably redundant, with numerous instances where the same points are reiterated unnecessarily. This redundancy detracts from the overall readability and coherence of the paper, making it challenging for readers to grasp the core contributions and findings effectively.

A glaring omission in the manuscript is the lack of a comprehensive Related Work section. The absence of this section makes it difficult for readers to contextualize the study within the current state-of-the-art literature. Without understanding how this work compares to previous studies, including its differences and similarities, it's challenging to appreciate its novelty and significance fully.

The exclusive reliance on GloVe embeddings for the analysis is another limitation. While GloVe is a widely used method for word representation, the paper would greatly benefit from a comparison with other embedding techniques. Incorporating a broader range of embeddings, especially more recent models such as contextualized embeddings that leverage masking strategies or generative models that utilize prompt strategies, could significantly strengthen the analysis. These models could offer insights into semantic relatedness that differ from those provided by GloVe, thus enriching the study's findings.

Furthermore, the paper could enhance its analysis by incorporating lexical resources like WordNet. Such resources would allow for a more nuanced exploration of semantic relations beyond mere co-occurrence frequencies. They could enable the examination of specific types of associations, such as hierarchical or antonym relationships, and help address the polysemy issue, which is overlooked in the current study.

Lastly, the study's scope, limited to just five cue words, raises questions about its statistical sufficiency and the ability to generalize its findings. A broader selection of words would likely provide a richer and more diverse dataset, potentially yielding more robust and comprehensive insights into semantic drift and word association dynamics.

In summary, while the paper tackles an intriguing topic, its impact is hindered by issues related to writing style, literature contextualization, methodological limitations, and the scope of analysis. Addressing these concerns would significantly improve the manuscript's clarity, depth, and relevance to the field.

Reviewer #2: The paper presents a keystroke study that tests whether semantic distance between a cue and provided associated terms varies more over time, or in other words, whether the most closely associated words are entered first by most participants (testing both distance to cue word and distance among each other). Other hypotheses tests whether the first associations are more similar across participants, whether the effect is stronger for concrete terms compared to abstract ones, and the effect of overall frequency. The experiment is carried out presenting five cues to 400+ participants with 4 out of 5 hypotheses confirmed with statistically significant results.

Though the paper is generally well-written and especially the presentation of results is nicely structured, I only fully understood the experimental set-up relatively late. I therefore recommend providing a very simple explanation of the task that participants carried out early on and explicitly mentioning what the quartiles are.

It is still not completely clear to me how the semantic drift investigated here relates to other forms of semantic drift. A more explicit explanation would strengthen the paper.

I'm listing a few typos below.

l 46-67 theories [..] has => have

l 64 -65 which has led ... => this sentence does not really work

l408 terms, would => terms would

6. PLOS authors have the option to publish the peer review history of their article (what does this mean?). If published, this will include your full peer review and any attached files.

Reviewer #1: No

Reviewer #2: No

---

## [Author Response · Author response to Decision Letter 0]

2 May 2024

Reviewer 1

1. Firstly, the writing style is notably redundant, with numerous instances where the same points are reiterated unnecessarily. This redundancy detracts from the overall readability and coherence of the paper, making it challenging for readers to grasp the core contributions and findings effectively. 

RESPONSE: We have attempted to address this with several deletions and simplifications. We have reduced our figure count to three, as most of the figures provided did not add new information to the tables that were already provided. Should any more examples have been missed, please don’t hesitate to cite a couple of examples and we will attempt to refine further. 

2. A glaring omission in the manuscript is the lack of a comprehensive Related Work section. 

RESPONSE: This has been amended now with a Related Work Section. This includes a reorganisation of the original intro and new paragraphs. 

3 and 4. The exclusive reliance on GloVe embeddings for the analysis is another limitation…Furthermore, the paper could enhance its analysis by incorporating lexical resources like WordNet. 

RESPONSE: Have now added BERT and WordNet analyses, and updated the methods, results, and discussions accordingly. 

5. Lastly, the study's scope, limited to just five cue words, raises questions about its statistical sufficiency and the ability to generalize its findings. A broader selection of words would likely provide a richer and more diverse dataset, potentially yielding more robust and comprehensive insights into semantic drift and word association dynamics.

RESPONSE: We have added a limitations section that highlights this. We have also made several references in the discussion to following this research up with a larger set of cues and responses.

Reviewer 2 

1. I therefore recommend providing a very simple explanation of the task that participants carried out early on and explicitly mentioning what the quartiles are.

RESPONSE: Added to the abstract.

2. It is still not completely clear to me how the semantic drift investigated here relates to other forms of semantic drift. A more explicit explanation would strengthen the paper. 

RESPONSE: As noted for Reviewer 1, this has been amended now with a Related Word Section This includes a reorganisation of the original intro and new paragraphs. 

3. Typos 

RESPONSE: Corrected

---

## [Decision Letter · Decision Letter 1]

27 May 2024

PONE-D-24-02236R1Keystrokes: A practical exploration of semantic drift in timed word association tasksPLOS ONE

Dear Dr. MacNiven,

Thank you for submitting your manuscript to PLOS ONE. After careful consideration, we feel that it has merit but does not fully meet PLOS ONE’s publication criteria as it currently stands. Therefore, we invite you to submit a revised version of the manuscript that addresses the points raised during the review process.

Many thanks for your attention to the reviewers' feedback.  The manuscript is greatly improved, and there are only two minor comments from R2 remaining that should be addressed prior to publication. Please submit a revision and response addressing them, and I will render an editorial decision without re-sending the manuscript to the reviewers.

We look forward to receiving your revised manuscript.

Kind regards,

Laura Morett

Academic Editor

PLOS ONE

Journal Requirements:

Reviewers' comments:

Reviewer's Responses to Questions

**Comments to the Author**

1. If the authors have adequately addressed your comments raised in a previous round of review and you feel that this manuscript is now acceptable for publication, you may indicate that here to bypass the “Comments to the Author” section, enter your conflict of interest statement in the “Confidential to Editor” section, and submit your "Accept" recommendation.

Reviewer #1: (No Response)

Reviewer #2: All comments have been addressed

2. Is the manuscript technically sound, and do the data support the conclusions?

Reviewer #1: (No Response)

Reviewer #2: Yes

3. Has the statistical analysis been performed appropriately and rigorously? 

Reviewer #1: (No Response)

Reviewer #2: I Don't Know

4. Have the authors made all data underlying the findings in their manuscript fully available?

Reviewer #1: (No Response)

Reviewer #2: Yes

5. Is the manuscript presented in an intelligible fashion and written in standard English?

Reviewer #1: (No Response)

Reviewer #2: Yes

6. Review Comments to the Author

Reviewer #1: The authors of the paper have appropriately updated the paper based on the reviewers' comments. The paper needs no further revision.

Reviewer #2: It is nice to see the improvement in this revised version. I now find the paper clear, nice to read without redundancies, the methodology has been strengthened and a nice first step towards exploring shifts in short periods of times (as stated in the papers conclusion). This paper thus addresses the main points of improvement in the review. I have two comments left for preparing camera ready:

First:

L411: `the moderate effect sizes we obtained for all pre-registered hypotheses (except H4)’ How about Hypothesis 2? Here, the null hypothesis could not be rejected. I suspect H2 was included since there were effects, but not strong enough to reject the null hypothesis, but as a reader this is confusing. I recommend taking up H2 in the `except’ together with H4, possibly making the difference between the two explicit.

Second:

I recommend adding a short paragraph to the introduction summing up the main purpose and contribution of the paper. Redundancy between an abstract and introduction are okay.

7. PLOS authors have the option to publish the peer review history of their article (what does this mean?). If published, this will include your full peer review and any attached files.

Reviewer #1: No

Reviewer #2: No

---

## [Author Response · Author response to Decision Letter 1]

30 May 2024

Team Response

Line 418: Changed to “Although we were unable to reject H2 (stronger effect sizes for abstract versus concrete terms), the moderate and significant effect sizes we obtained for H1, H3 and H5a and b, may be regarded as promising.” 

Line 71: Added a new paragraph “The present study aims to explore this phenomenon in the context of short, free-word-associations tasks. In this study, we seek to explore to what extent meaning might evolve as we engage, cognitively, with a cue (or idea), and how might the process of imagining and remembering, change how far we veer away from common, established associations to personally relevant and meaningful simulations? Importantly, can such shifts in meaning be detected in a sub-minute word association task? And if so, how might that inform the design and analysis of future studies that include or even revolve around such free-word-associations?”

---

## [Editor Report · Decision Letter 2]

2 Jun 2024

Keystrokes: A practical exploration of semantic drift in timed word association tasks

PONE-D-24-02236R2

Dear Dr. MacNiven,

We’re pleased to inform you that your manuscript has been judged scientifically suitable for publication and will be formally accepted for publication once it meets all outstanding technical requirements.

Kind regards,

Laura Morett

Academic Editor

PLOS ONE

Additional Editor Comments (optional):

Many thanks for your revisions in response to the remaining points. I'm pleased to accept this manuscript for publication at this time.

---

## [Editor Report · Acceptance letter]

21 Jun 2024

PONE-D-24-02236R2 

PLOS ONE

Dear Dr. MacNiven, 

I'm pleased to inform you that your manuscript has been deemed suitable for publication in PLOS ONE. Congratulations! Your manuscript is now being handed over to our production team.

Kind regards, 

on behalf of

Dr. Laura Morett 

Academic Editor

PLOS ONE